

# Aerodynamic forces and flows of the full and partial clap-fling motions in insects

Xin Cheng and Mao Sun

Institute of Fluid Mechanics, Beijing University of Aeronautics and Astronautics, Beijing, China

## ABSTRACT

Most of the previous studies on Weis-Fogh clap-fling mechanism have focused on the vortex structures and velocity fields. Detailed pressure distribution results are provided for the first time in this study to reveal the differences between the full and the partial clap-fling motions. The two motions are studied by numerically solving the Navier–Stokes equations in moving overset grids. The Reynolds number is set to 20, relevant to the tiny flying insects. The following has been shown: (1) During the clap phase, the wings clap together and create a high pressure region in the closing gap between wings, greatly increasing the positive pressure on the lower surface of wing, while pressure on the upper surface is almost unchanged by the interaction; during the fling phase, the wings fling apart and create a low pressure region in the opening gap between wings, greatly increasing the suction pressure on the upper surface of wing, while pressure on the lower surface is almost unchanged by the interaction; (2) The interference effect between wings is most severe at the end of clap phase and the start of the fling phase: two sharp force peaks (8–9 times larger than that of the one-winged case) are generated. But the total force peaks are manifested mostly as drag and barely as lift of the wing, owing to the vertical orientation of the wing section; (3) The wing–wing interaction effect in the partial clap-fling case is much weaker than that in the full clap-fling case, avoiding the generation of huge drag. Compared with a single wing flapping with the same motion, mean lift in the partial case is enhanced by 12% without suffering any efficiency degradation, indicating that partial clap-fling is a more practical choice for tiny insects to employ.

# INTRODUCTION

The average wing length for an insect is about 3–4 mm (*Dudley, 2002*). Most of the previous studies, however, have focused on insects of relatively large size (*Walker, Thomas & Taylor, 2009*; *Cheng, Deng & Hedrick, 2011*; *Sun & Lan, 2004*; *Dudley & Ellington, 1990*; *Liu & Sun, 2008*; *Mou, Liu & Sun, 2011*; *Meng & Sun, 2015*; *Lua et al., 2016*; *Lu et al., 2016*). Tiny insects have a large population quantity and are of significant ecological and agricultural importance (*Terry, 2001*; *Crespi, Carmean & Chapman, 1997*; *Austin & Dowton, 2000*), but their mechanics of flight have been much less explored. Tiny insects fly at Reynolds number (*Re*) below 60 where viscous effects are significant (*Weis-Fogh, 1973*; *Santhanakrishnan et al., 2014*; *Cheng & Sun, 2016*). Previous numerical results of flapping wings have shown

Corresponding author
Xin Cheng, x.cheng@buaa.edu.cn

that the lift-to-drag ratio decreases greatly when $Re$ is below $\sim$100 (*Wang, 2000*; *Miller & Peskin, 2004*; *Wu & Sun, 2004*). With the challenges associated with generating lift at such low $Re$, tiny insects must employ additional flight strategies to enhance lift, such as wing interference and wing flexibility. The most well-known example of beneficial interaction is the "clap and fling" mechanism proposed by *Weis-Fogh (1973)* for the small wasp *Encarsia formosa* in hovering. Towards the end of the upstroke, the wings clap together by rotating about their leading edges; then, at the beginning of the subsequent downstroke, the wings fling apart by rotating about their trailing edges.

Later observations of tiny insects' flight suggested that the clap and fling is a common phenomenon at this scale; for example, the greenhouse white-fly *Trialeurodes vaoparium* (*Weis-Fogh, 1975*), thrips (*Santhanakrishnan et al., 2014*; *Ellington, 1984a*) and parasitoid wasp species *Muscidifurax raptor* and *Nasonia vitripennis* (*Miller & Peskin, 2009*). It is highly surprising that a marine mollusk, *Limacina helicina* also performs a "near fling" maneuver at stroke-reversal to augment lift ($Re = 40$–90) (*Murphy et al., 2016*). Larger insects rarely use clap and fling mechanism in free flight, with few exceptions, such as some moths in forward flight (*Ellington, 1984a*), some butterflies in take-off flight (*Sunada & Kawachi, 1993*), and some locusts in climbing flight (*Cooter & Baker, 1977*). The small size and high wing beat frequency of tiny insects pose difficulty for video recording and the aforementioned tiny insects were all filmed with only one or two high-speed cameras. Consequently, no complete quantitative description of the clap and fling motion is available. Some progress in this field is made by *Cheng & Sun (2016)* who obtained detailed wing kinematics (including positional angle, deviation angle and pitch angle) of a small fly, vegetable leafminer *Liriomyza sativae*, hovering at $Re \approx 40$. At dorsal stroke-reversal, the small fly has a partial clap-fling motion, which is a subtle variation of full clap-fling motion described for *Encarsia formosa*. More specifically, in the full clap-fling motion, both wings are parallel in close proximity along their entire surface at the dorsal stroke-reversal; in the partial clap-fling motion, only the outer parts of the wings are in close proximity and the wing roots are farther apart than the wing tips.

Inspired by Weis-Fogh's 1973 paper (*Weis-Fogh, 1973*), the lift-augmenting mechanism of the full clap-fling motion was widely studied by many researchers using theoretical methods (*Lighthill, 1973*; *Edwards & Cheng, 1982*; *Ellington, 1984b*; *Wu & Chen, 1984*; *Kolomenskiy et al., 2011a*), physical models (*Maxworthy, 1979*; *Spedding & Maxworthy, 1986*; *Lehmann & Pick, 2007*) and numerical simulations (*Santhanakrishnan et al., 2014*; *Miller & Peskin, 2009*; *Ro & Tsutahara, 1997*; *Sun & Yu, 2003*; *Miller & Peskin, 2005*; *Sun & Yu, 2006*; *Kolomenskiy et al., 2010*; *Kolomenskiy et al., 2011b*; *Arora et al., 2014*; *Jones et al., 2015*). In contrast, the partial clap-fling motion was thought to be just a variant of the full clap-fling motion and received much less attention (*Cheng & Sun, 2016*; *Lehmann, Sane & Dickinson, 2005*). All previous works supported the idea that the wing–wing interaction during clap-fling motion augments lift. For wings performing full clap-fling motion, the mean lift coefficient can be significantly enhanced by about 20%–70% (*Miller & Peskin, 2005*; *Sun & Yu, 2006*), depended on different configurations of wing kinematic models; for wings performing partial clap-fling motion, the mean lift coefficient can be increased

by about 7%–9% (*Cheng & Sun, 2016*; *Lehmann, Sane & Dickinson, 2005*), indicating that partial clap-fling is somewhat less effective than the full clap-fling in enhancing lift.

However, the wing interference effects on drag force performances have been largely ignored in most previous studies. Based on the simulation results of the previous two- and three-dimensional computational works (*Miller & Peskin, 2009*; *Miller & Peskin, 2005*; *Sun & Yu, 2006*), extremely large drag forces are produced in the full clap-fling cases at Reynolds numbers of tiny insects ($Re \approx 10$). For certain configurations, the drag required to fling the wings apart can be an order of magnitude larger than that required by a single wing with the same motion (*Miller & Peskin, 2009*). If so, the aerodynamic torque around the axis of azimuthal rotation, which is due to the drag force, will become too large for the wing hinge to support. Though can be reduced by about 50% with adding wing flexibility (*Miller & Peskin, 2009*) or increasing initial distance between wings (*Sun & Yu, 2006*), the drag forces and relevant aerodynamic torques are still several times as large as those of the single wing. Using the measured wing motion data, *Cheng & Sun (2016)* calculated the aerodynamic forces of a tiny insect vegetable leafminer (it has a partial clap-fling motion at the dorsal stroke reversal). Their results showed that the drag forces with aerodynamic interaction are comparable to those without aerodynamic interaction, but they did not analyze the difference in detail. *Lehmann, Sane & Dickinson (2005)* studied the partial clap-fling employing robotic fruit-fly wings ($Re \approx 100$) and showed similar results. Therefore, it is worthwhile to compare the aerodynamic performances between the full and the partial clap-fling motions in 3D circumstances and find out what causes the drag differences between the two cases.

The key feature of the partial clap-fling is that the wing separation varies along the wing span during clap-fling phase, it is predictable that the interference effect between wings decreases from wing tip to wing root. Given the radial dependency of flow down the span and the uniform application of wing separation in full clap-fling, it is interesting to figure out how the wing interference effect varies in the wing span direction with full clap and fling. Moreover, highlighting lift force only or discussing lift and drag force performances separately gives incomplete understanding of clap-fling mechanism. For the flapping wing in this study, the $Re$ is still high enough to assume that the aerodynamic force is almost normal to the wing surface and the lift and drag forces are components of the normal force perpendicular to and parallel to the stroke plane, respectively. It would be very helpful to firstly identify the underlying wing–wing interaction effect accounts for the total force difference, then the lift and drag force differences can be calculated depending on the orientation of the wing.

In the present study, we investigate two idealized motions, i.e., the 3D full clap-fling motion (constructed based on *Sun & Yu, 2006*) and the 3D partial clap-fling motion (constructed based on *Cheng & Sun, 2016*). The aerodynamic forces and flow structure around model wings are computed by solving the Navier–Stokes equation numerically. The $Re$ is set to 20. The interference effects are identified by comparing the results between single-winged and double-winged cases with the same wing motion. For the first time, we provide detailed pressure distribution results around wings during the complete clap-fling

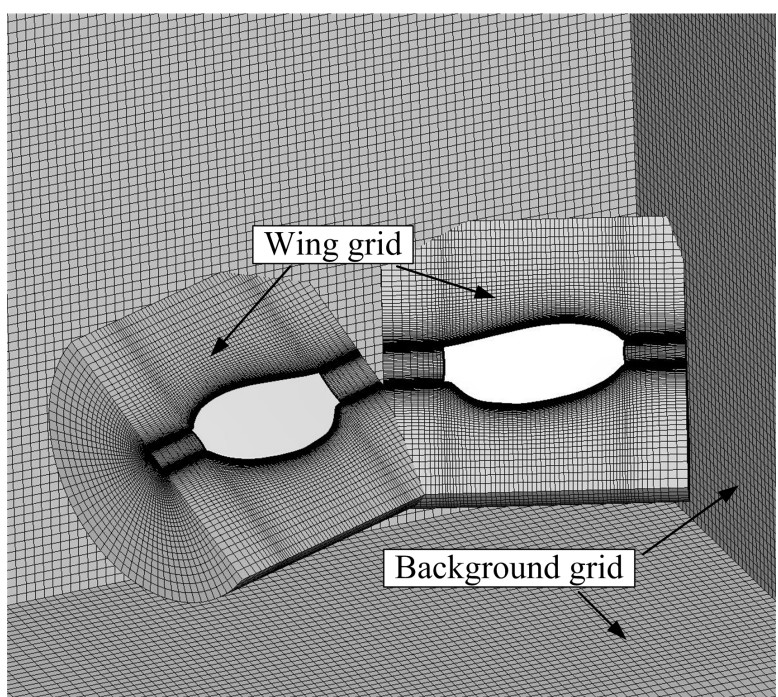

**Figure 1** **Model wings and portions of the computational overset grids.**

motion. How the "clapping together" and "flinging apart" of wings influence the fluids nearby is revealed by analyzing the flow fields and pressure distributions.

## MATERIALS AND METHODS

### Model wings and wing motions

To compare the difference between the full and partial clap-fling motions, only one wing shape should be used in our simulation. As mentioned above, the full clap-fling motion constructed based on *Sun & Yu (2006)* and the partial clap-fling motion constructed based on *Cheng & Sun (2016)* are used in the present study. We might either use the wing shape in *Sun & Yu (2006)* or that in *Cheng & Sun (2016)*. Here, we use the wing planform of the *vegetable leafminer* (VL1) in *Cheng & Sun (2016)*. The model wings and portions of computational grids are shown in Fig. 1. The wing section is assumed to be rigid flat plate. For small insects like fruitfly (*Meng & Sun, 2015*) and *vegetable leafminer* (*Cheng & Sun, 2016*), their wings are small and relatively stiff so that the wing deformation is much smaller when compared with large insects like hawkmoth (*Lua et al., 2010*) The aspect ratio ($AR$), i.e., the ratio of the wing length ($R$) to the mean chord length of the wing ($c$), is 3.43. The radius of gyration ($r_2$) of the model wing is $0.59R$; the mean flapping velocity at the span location $r_2$ of the wing is defined as the reference velocity $U$ (*Lua, Lim & Yeo, 2014*) (defined below).

Because the left wing is always the mirror image of the right wing during its motion, we only describe the motion of the right wing. To clearly describe the wing motion, two coordinate systems are introduced (Fig. 2A). Let $OXYZ$ be an inertial frame with its origin

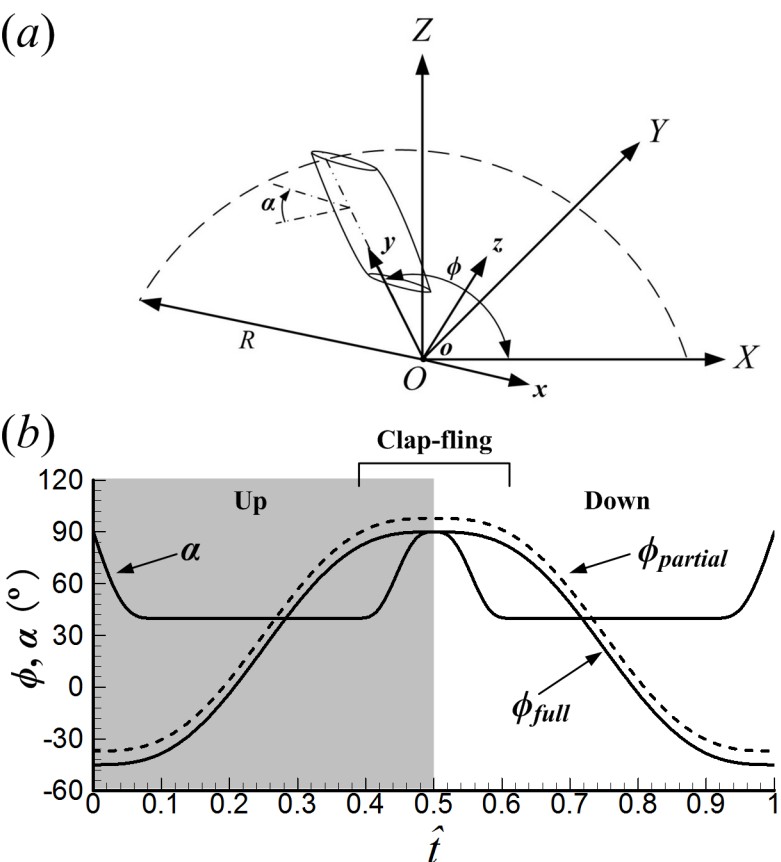

**Figure 2** (A) Sketches of the reference frames and wing motion. (B) Time history of positional angle ($\phi$) and angle of attack ($\alpha$) in one cycle ($\phi_{full}$ represents the positional angle of the full clap-fling motion; $\phi_{partial}$ represents the positional angle of the partial clap-fling motion).

at the wing root and $oxyz$ be a non-inertial system with the same origin. For $OXYZ$, $X$-axis and $Y$-axis form the horizontal plane (parallel to the stroke plane), and the $Z$-axis is vertical. $oxyz$ is a frame fixed on the wing, with its $x$-axis parallel to the wing chord line and the $y$-axis along the wing span. Thus, the $z$-axis is normal to the wing surface. Based on the data given by *Weis-Fogh (1973)* and *Ellington (1975)*, the full clap-fling motion is constructed as follows. The stroke cycle begins with the upstroke. Initially, the chord line of the wing is in vertical direction; at the start of the upstroke, the wing rotates around the flip axis ($y$-axis); after the flip, the wing sweeps (rotates azimuthally around the $Z$-axis, referred as "translation") from the ventral to the dorsal side of the body; near the end of the upstroke, the wing rotates about its leading edge (the clap) and becomes vertical at the end of the upstroke; then the wing rotates about its trailing edge (the fling) at the beginning of the downstroke; following the fling, the wing sweeps from the dorsal to the ventral side of the body; near the end of the downstroke, the wing rotates about the flip axis until its chord line is in vertical direction again.

Let $\phi$ denote the positional angle of translation (Fig. 2A). In the present study, the time variation of $\phi$ is approximated by a simple harmonic function:

$$\phi = \overline{\phi} - 0.5\,\Phi\cos(2\pi\tau/\tau_c) \quad 0 \leq \tau \leq \tau_c \tag{1}$$

where $\overline{\phi}$ is the mean stroke angle, $\Phi$ the stroke amplitude, $\tau$ the non-dimensional time, and $\tau_c$ the non-dimensional period of the flapping cycle. $\overline{\phi}$, $\Phi$, $\tau$ and $\tau_c$ are defined as follows: $\overline{\phi} = (\phi_{max} + \phi_{min})/2$, $\Phi = \phi_{max} - \phi_{min}$, $\tau = tU/c$, and $\tau_c = U/cn$, where $\phi_{max}$ is the maximum value of $\phi$, $\phi_{min}$ the minimum value of $\phi$, $t$ the real time and $n$ the wingbeat frequency. The reference velocity $U$ is defined as $U = 2\,\Phi n r_2$.

The angle of attack of the wing ($\alpha$), i.e., the angle between the chord line and stroke plane (Fig. 2A), is assumed to be a constant value ($\alpha_m$) in the mid-portion of a half-stroke except at stroke reversal (we call an up- or downstroke as a half-stroke). The ventral stroke reversal is a normal flip employed by most insects while the dorsal stroke reversal ("clap and fling") is a special flip with different $\alpha$ variation. The time variation of $\alpha$ during the flip rotation at the start of upstroke, the clap, the fling, and flip rotation at the end of downstroke are given as, respectively,

$$\alpha = \frac{\pi}{2} - \left(\frac{\pi}{2} - \alpha_m\right)\sin\left(\frac{\pi\tau}{2\Delta\tau_r}\right) \quad 0 \leq \tau \leq \tau_r \tag{2}$$

$$\alpha = \alpha_m + 0.5\left(\frac{\pi}{5} - \alpha_m\right)\left(1 - \cos\left(\frac{\pi(\tau - 0.5\tau_c + \Delta\tau_{cl})}{\Delta\tau_{cl}}\right)\right) \quad 0.5\tau_c - \Delta\tau_{cl} \leq \tau \leq 0.5\tau_c \tag{3}$$

$$\alpha = \frac{\pi}{2} 0.5\left(\frac{\pi}{2} - \alpha_m\right)\left(1 - \cos\left(\frac{\pi(\tau - 0.5\tau_c)}{\Delta\tau_f}\right)\right) \quad 0.5\tau_c \leq \tau \leq 0.5\tau_c + \Delta\tau_f \tag{4}$$

$$\alpha = \alpha_m + \left(\frac{\pi}{2} - \alpha_m\right)\left(1 - \cos\left(\frac{\pi(\tau - \tau_c + \Delta\tau_r)}{2\Delta\tau_r}\right)\right) \quad \tau_c - \Delta\tau_r \leq \tau \leq \tau_c \tag{5}$$

where $\Delta\tau_r$ is the flip rotation, $\Delta\tau_{cl}$ the clap duration and $\Delta\tau_f$ the fling duration. Here, $\tau_r$ is the time at which the flip rotation of the upstroke ends and $0.5\tau_c$ is the time at which the upstroke ends.

The Reynolds number ($Re$), which appears in the non-dimensional Navier–Stokes equations, is defined as $Re = cU/\nu$, where $\nu$ is the kinematic viscosity of the air. $Re$ is set to 20 in this paper. Based on the morphological data of the model wing used in the present study, the non-dimensional period is computed as follows: $\tau_c = U/cn = 2\,\Phi r_2/c = 9.54$. The idealized full clap-fling motion used in this paper is similar to that used by *Sun & Yu (2006)*. We set $\phi_{max} = 90°$, $\phi_{min} = -45°$, $\alpha_m = 40°$, $\Delta\tau_r = 0.085\tau_c$, $\Delta\tau_{cl} = \Delta\tau_f = 0.11\tau_c$. To clearly describe the wing motion, we express the time during a cycle as a non-dimensional parameter $\hat{t}$: $\hat{t} = 0$ at the start of an upstroke and $\hat{t} = 1$ at the end of the subsequent downstroke. The time courses of $\phi$ and $\alpha$ of the right wing in one cycle is shown in Fig. 2B. For the full clap-fling motion, the two wings are in the closest proximity at the end of upstroke (Fig. 3). Let $d$ denotes the distance between wings at this instance (see Fig. 3, $\hat{t} = 0.5$). In our previous study (*Cheng & Sun, 2016*), the minimum distance between wing tips of VL1 was about $0.11c$, very close to $d = 0.1c$, which had been chosen in several

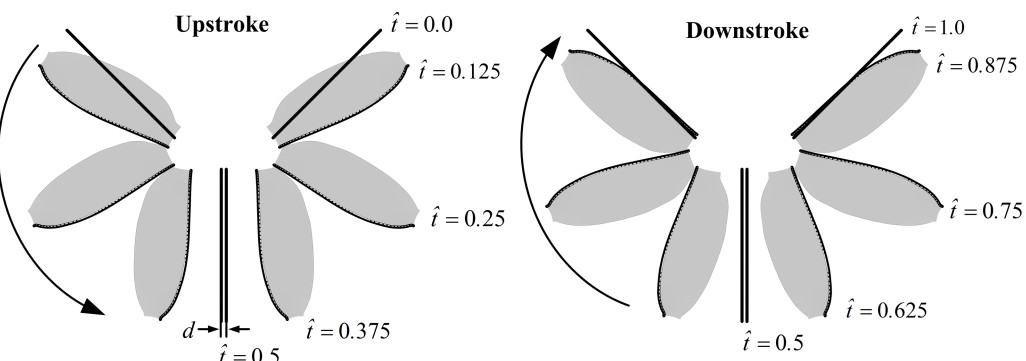

**Figure 3** Schematic of the wing pair's motion (3D, top view) for the full clap-fling in a complete stroke cycle.

previous numerical studies (*Sun & Yu, 2003*; *Miller & Peskin, 2005*; *Sun & Yu, 2006*; *Arora et al., 2014*). Therefore, we specify $d = 0.1c$ in this study.

The partial clap-fling motion is constructed on the basis of the full clap-fling motion described above by modifying two parameters: the distance between wing roots and the positional angle ($\phi$). More specifically, we obtain the partial clap-fling motion by enlarging the distance between wing roots and introducing an angular excursion of $+7.9°$ to $\phi$ in the full clap-fling motion (the two values are decided based on the wing kinematics of VL1 in our previous study (*Cheng & Sun, 2016*)) (Figs. 2B and Fig. 4). As a result, in the partial clap-fling motion, only the outer parts of the wings are in close proximity during clap and fling, the distance between wing tips at $\hat{t} = 0.5$ is also $d = 0.1c$. For comparison, a single wing performing identical motion as that of a wing of the wing pair performing the full clap-fling motion is also considered. In this study, only one flapping cycle is simulated.

## Flow solution method and aerodynamic forces

The flows around the wings are computed numerically by solving the Navier–Stokes equations. In solving the Navier–Stokes equations, moving overset grids are used because the left and right wings are in close proximity during clap-fling motion, and there can be strong aerodynamic interactions between the wings. The numerical method is the same as that used by Sun et al. in several previous studies (*Cheng & Sun, 2016*; *Sun & Yu, 2006*). A description of it is given in the Supplemental Information 1; the computational grids and grid resolution tests are also discussed there.

Solving the Navier–Stokes equations gives the fluid velocity components and pressure at discretized grid points for each time step. Here, the total aerodynamic force of a wing ($F$) is computed by integrating the pressure and viscous stress over the wing surface. Lift ($L$) is defined as the vertical component of the total aerodynamic force ($F$) and drag ($D$) as the component that is in the horizontal plane ($OXY$ plane) (note that the $OXY$ plane is coincide with the stroke plane). Normal force ($N$) is the force normal to the wing surface and calculated by only integrating the pressure difference across the wing surface. For the flapping wing (thin wing operating at high angle of attack and the flow being separated), the normal force ($N$) is almost identical to the total force ($F$) and the lift ($L$) and drag

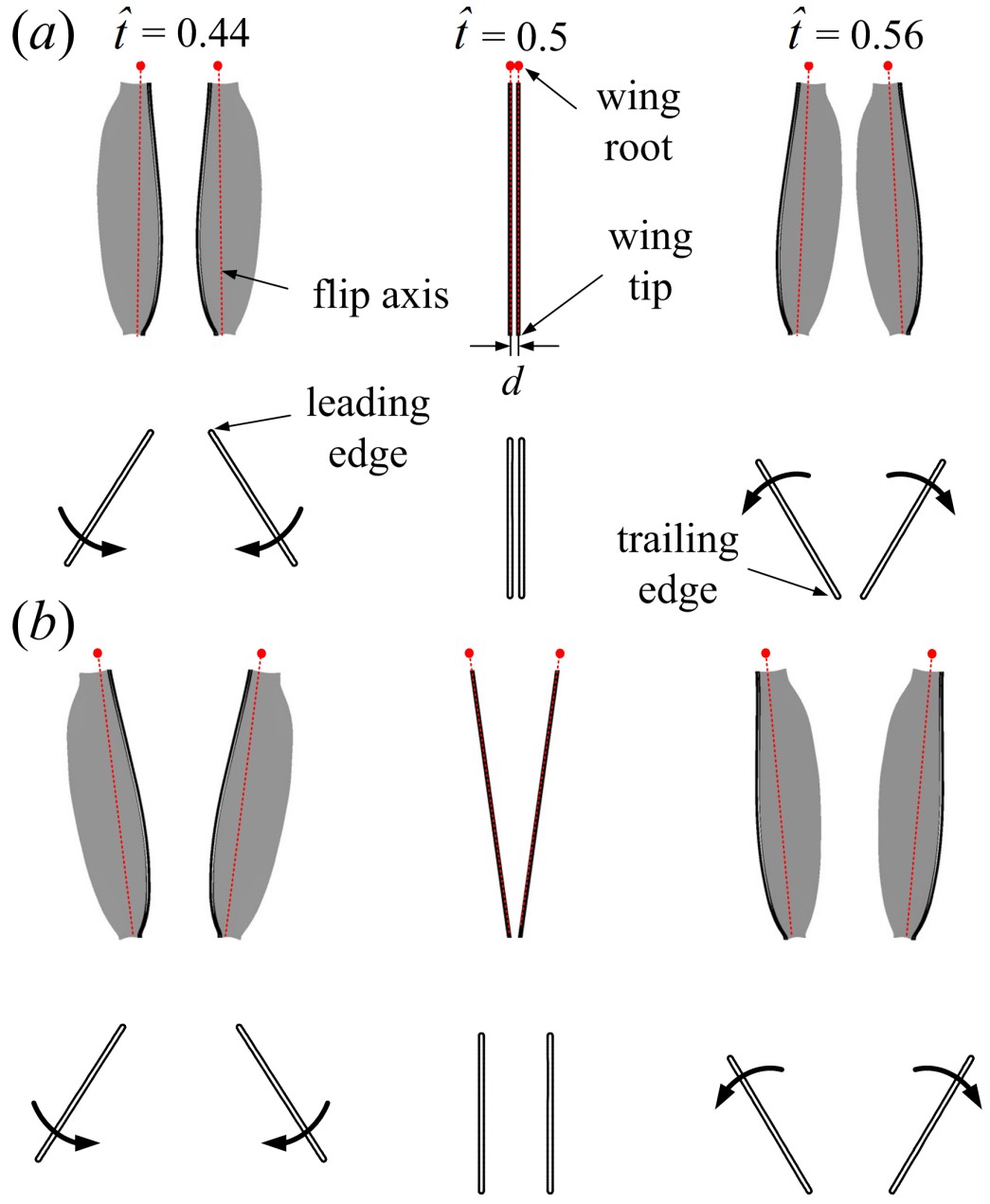

**Figure 4** Motions of the wing pair (3D, top view) and the corresponding wing sections (2D, side view) at half-wing length for the full (A) and the partial (B) clap-fling at three time instances ($\hat{t} = 0.44$, $\hat{t} = 0.5$ and $\hat{t} = 0.56$) during dorsal stroke reversal.

($D$) can be approximated by the components of the normal force ($N$) normal to and parallel to the stroke plane, respectively. The corresponding coefficients of the above forces (denoted as $C_F$, $C_N$, $C_L$ and $C_D$, respectively) are defined as follows: $C_F = F/0.5\rho U^2 S$, $C_N = N/0.5\rho U^2 S$, $C_L = L/0.5\rho U^2 S$ and $C_D = D/0.5\rho U^2 S$, where $\rho$ is the fluid density and $S$ is the wing area.

## RESULTS AND DISCUSSION

Using the wing motion data and numerical methods described in the 'Materials and Methods' section, flows and aerodynamic forces of the full and the partial clap-fling cases were computed at $Re = 20$, and the results were compared between the two cases and also with the corresponding single-winged case. For convenience, the full clap-fling case, the partial clap-fling case and the single-winged case are called FCF case, PCF case and SW case, respectively in the later discussions.

### Aerodynamic force differences

As aforementioned, normal force ($N$) is the total force perpendicular to the wing surface, lift ($L$) can be approximated as the component of $N$ in the vertical direction ($Z$ direction) and drag ($D$) as the component in the stroke plane and normal to the wing span. When the chord line of the wing is parallel to the stroke plane, i.e., $\alpha = 0°$, the total normal force reorients in the lift direction; when the chord line is in the vertical direction, i.e., $\alpha = 90°$, the total normal force reorients in the drag direction. Figure 5 gives the time courses of $C_N$, $C_L$ and $C_D$ in one cycle for the three different cases (the time histories of angular velocity $\dot{\phi}$ and $\dot{\alpha}$ are also given in the figure). It is seen that the interaction effect between wings is mostly restricted to the clap-fling phase ($\hat{t} = 0.39$–$0.61$) and decays rapidly once the wings move apart from each other. Moreover, $C_N$ of the FCF case is much larger than that of the PCF case in the mid-portion of the clap-fling phase ($\hat{t} \approx 0.45$–$0.55$, Fig. 5A). And it should be noted that, the large $C_N$ obtained during the clap-fling phase in the FCF case is manifested mostly as $C_D$ rather than $C_L$ (compare the magnitude of $C_L$ and $C_D$ in Figs. 5C and 5D), especially at the end of the clap phase ($\hat{t} = 0.48$) and the start of the fling phase ($\hat{t} = 0.52$), the two sharp $C_N$ peaks are nearly equal to the two sharp $C_D$ peaks. This is resulted from the orientation of the wing. In the mid-portion of dorsal stroke reversal ($\hat{t} \approx 045$–$0.55$), the pitch angle of attack ($\alpha$) is large (see Fig. 2B) and the wing plane is more vertical than that in the translation phase, so the total normal force reorients more in the drag direction and less in the lift direction. In addition to the sharp $C_N$ peak at $\hat{t} = 0.48$ obtained only in the FCF case, there is another $C_N$ peak at the start of clap phase ($\hat{t} \approx 0.43$) which is obtained in all the three cases. This $C_N$ peak is produced by the "fast-pitching-up rotation" mechanism, proposed by *Sun & Tang (2002)*. In the early portion of clap ($\hat{t} = 0.39$–$0.45$, see $\dot{\alpha}$ in Fig. 5A) while translating, the wing performs a fast pitching-up motion, generating strong vorticity in a short time and hence large aerodynamic force (*Meng & Sun, 2015*; *Sun & Tang, 2002*).

The two drag peaks of the FCF case are about 8–9 times as large as those in the SW case at the same instances and cause a serious aerodynamic problem. To sweep the wing back and forth in the stroke plane, the insect needs to overcome the aerodynamic torque and inertial torque for translation (around the $Z$-axis) at the wing root, while the aerodynamic torque is directly resulted from the drag force. In the FCF case, the aerodynamic torques for translation at $\hat{t} = 0.48$ and $\hat{t} = 0.52$ also become 8–9 times larger than those in the SW case, such large torque will be too large for the wing hinge to support. In the PCF case, in contrast, the wing–wing interaction effect is more moderate than that in the FCF case,

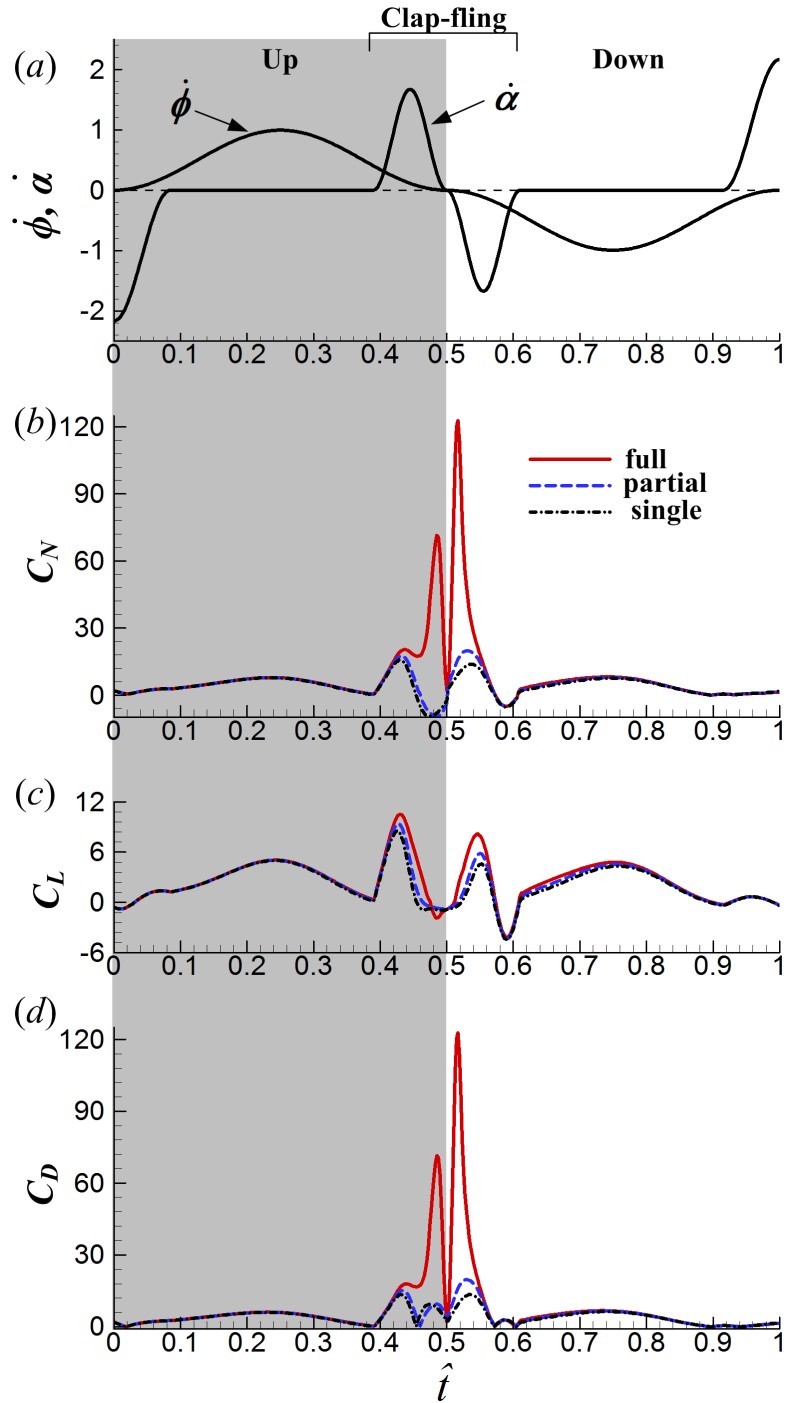

**Figure 5  Time courses of the wing motion and force coefficients in one cycle for the three different cases:** (A) non-dimensional angular velocity of rotation ($\dot{\alpha}$) and translation ($\dot{\phi}$); (B) the normal force coefficient; (C) the lift coefficient; (D) the drag force coefficient.

avoiding the generation of huge drag; there are only small quantitative differences in force coefficients between the PCF case and the SW case (Figs. 5B–5D).

To evaluate the influence of wing–wing interaction, the force differences between the one and two-winged cases were calculated (Fig. 6). As shown in Fig. 6B, the interference effect makes the average $C_L$ during the clap-fling phase ($\hat{t} = 0.39$–$0.61$) 2.4 times and 1.5 times as large as that of the SW case in the FCF case and in the PCF case, respectively. Later in the downstroke after the fling phase ($\hat{t} = 0.61$–$1.0$), the lift-enhancing effect reduces rapidly but still slightly increases $C_L$: average $C_L$ during this period increases 20% for the FCF case and 10% for the PCF case, compared with that of the SW case. Averaging $C_L$ and $C_D$ in one cycle ($\hat{t} = 0.0$–$1.0$) gives the mean lift coefficient ($\overline{C}_L$) and mean drag coefficient ($\overline{C}_D$) respectively. In the FCF case, $\overline{C}_L$ is increased by about 28% by the interaction, while $\overline{C}_D$ is approximately doubled compared to the SW case. *Sun & Yu (2006)* studied the 3D full clap-fling motion similar to that in the present work ($Re = 15$). Their computational results showed that $\overline{C}_L$ could be augmented by 20% and $\overline{C}_D$ be increased by 52%, comparable to our results. In the PCF case, $\overline{C}_L$ is modestly enhanced, by about 12%, and is only increased by about 10%. By employing robotic fruitfly wings ($Re \approx 100$), *Lehmann, Sane & Dickinson (2005)* studied the partial clap-fling motion similar to that in this paper. Their measurements showed that $\overline{C}_L$ could be augmented by about 9%, similar to our results. If the aerodynamic efficiency is defined as the lift-to-drag ratio ($\overline{C}_L/\overline{C}_D$), the full clap-fling motion appears to be rather inefficient. The partial clap-fling motion, however, can contribute more than 10% additional lift at tiny insect scale without suffering efficiency degradation. In this view, the partial clap-fling motion, rather than the full clap-fling motion, is a more practical choice for tiny insets to employ.

One interesting point to be noted in Fig. 6 is that, $C_D$ in the PCF case is even reduced somehow during the clap phase compared to the SW case ($\hat{t} = 0.46$–$0.48$ in Fig. 6C), which would increase power efficiency of the PCF case.

## Interference effects between wings in the clap phase

To reveal the underlying fluid mechanics of the above aerodynamic effects of wing interaction in the FCF and PCF cases, the velocity fields and surface pressure distributions at several instances are investigated (see Fig. 6A, $\hat{t} = 0.44$, $0.45$, $0.47$ and $0.48$ in the clap phase; $\hat{t} = 0.52$, $0.53$ and $0.55$ in the fling phase and $\hat{t} = 0.62$, $0.64$ and $0.66$ in the subsequent translation after fling). How the underlying clap-fling mechanism influences the total force difference is discussed in the following.

First, we examine the clap phase ($\hat{t} = 0.39$–$0.50$). Figure 7 gives the corresponding sectional normal force distribution of the FCF, PCF and SW cases along the wing span at $\hat{t} = 0.44$, $0.45$, $0.47$ and $0.48$ ($C_n$ denotes the coefficient of the sectional normal force and $r$ is the radial distance from the wing root). Figure 8 gives the surface pressure distribution of the three cases at several spanwise positions at the same time instants ($C_P$ denotes the pressure coefficient and is defined as $C_P = (p - p_\infty)/0.5\rho U^2$; solid and broken lines indicate the pressure distribution on the lower surface and the upper surface, respectively; $c_I$ denotes the local chord length of the wing at certain spanwise position). Note that the area enclosed by the curves representing the pressure coefficient ($C_P$) in Fig. 8 is the

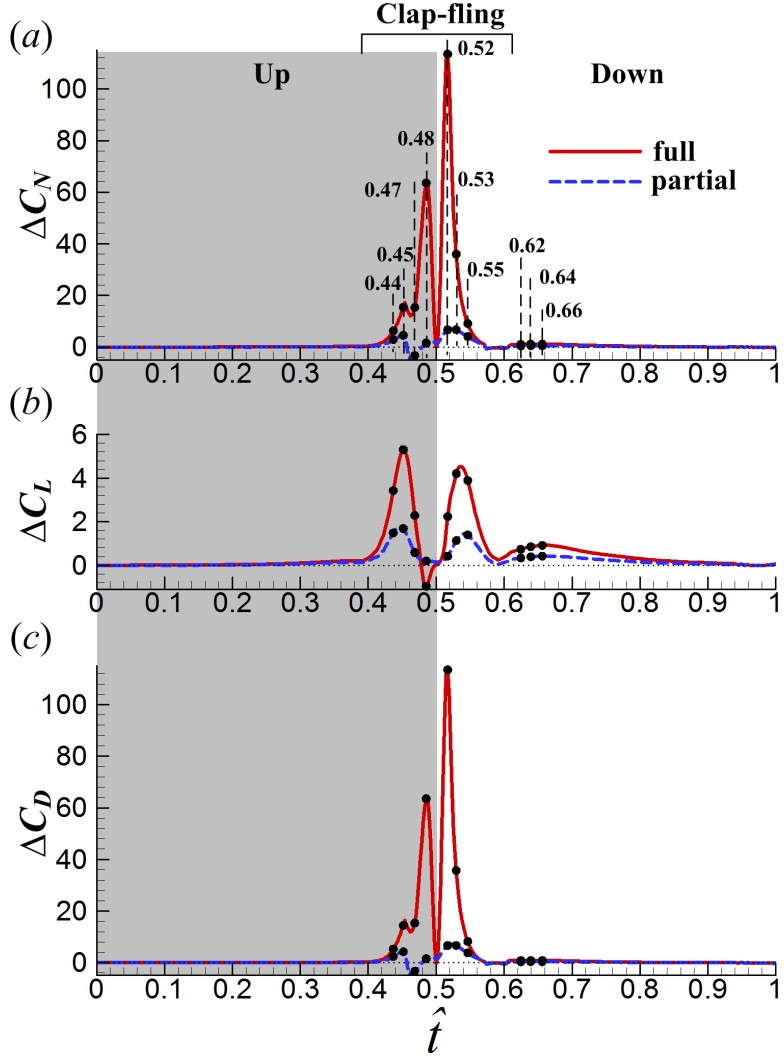

**Figure 6** Time courses of difference in the normal force coefficient ($\Delta C_N$) (A), lift coefficient ($\Delta C_L$) (B) and drag coefficient ($\Delta C_D$) (C) in one cycle between the one and two-winged cases (the ten small black dots indicate the instances at which detailed flowfield results were discussed).

non-dimensional sectional normal force ($C_n$) at the specified spanwise position in Fig. 7. During the clap phase ($\hat{t} = 0.39–0.50$), the positional angle ($\phi$) is approximately constant (Fig. 2B) and the wing quickly rotates about the leading edge, $\alpha$ increasing from 40° to 90° in a short period (see $\alpha$ and in $\dot{\alpha}$ Fig. 2B and Fig. 5A, respectively).

As seen in Fig. 7, $C_n$ of the FCF case is significantly larger than that of the SW case and the $C_n$ difference increases greatly with time. This is because, in the FCF case, the two wings are in close proximity. The width of the gap between wings is very small and does not change along the spanwise direction of wing, resulting in strong interaction effect between wings. In contrast, the $C_n$ difference between the PCF and the SW cases increases smoothly from wing root to wing tip: $C_n$ of the PCF case is almost the same as that of the SW case at wing root and is almost the same as that of the FCF case at wing tip. This is because, in the

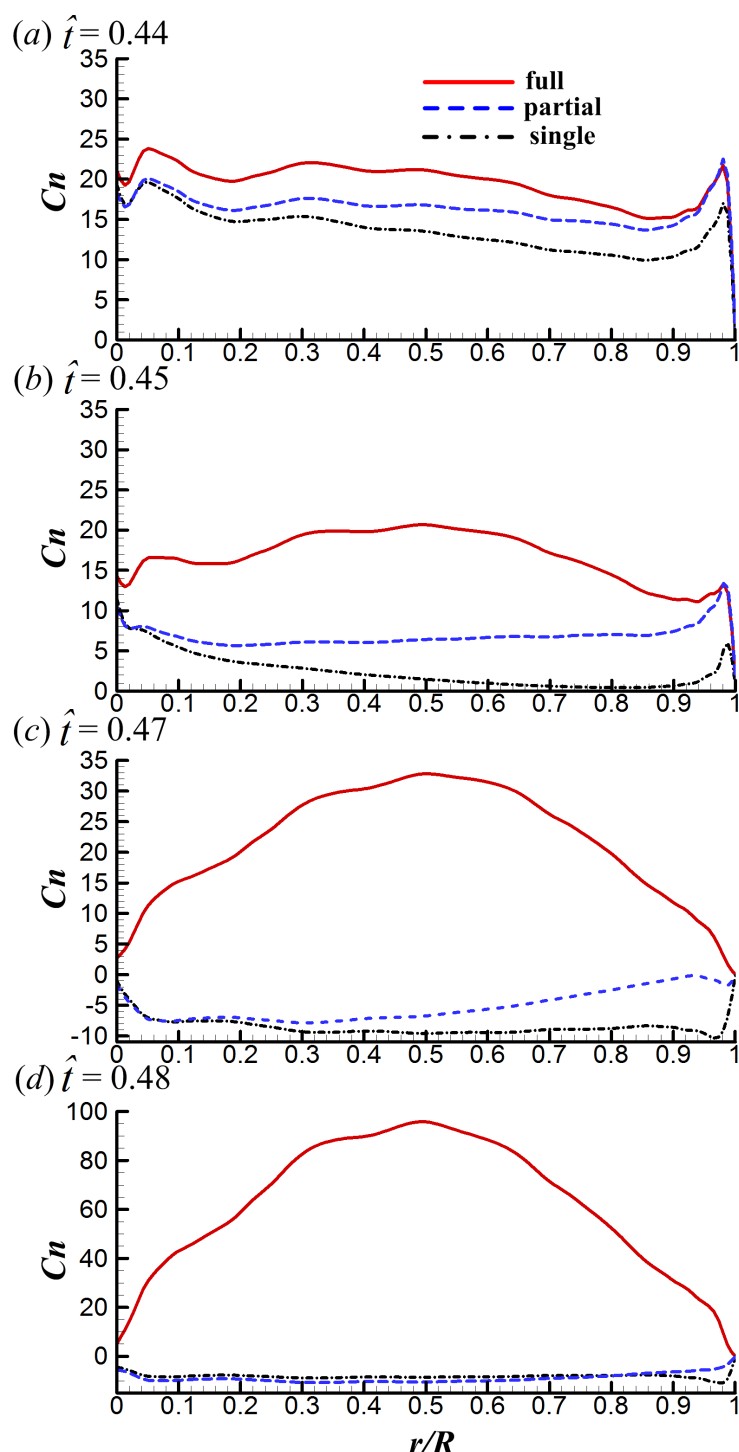

**Figure 7** The spanwise distributions of the sectional normal force of the three cases considered at $\hat{t} =$ 0.44, 0.45, 0.47 and 0.48 in the clap phase.

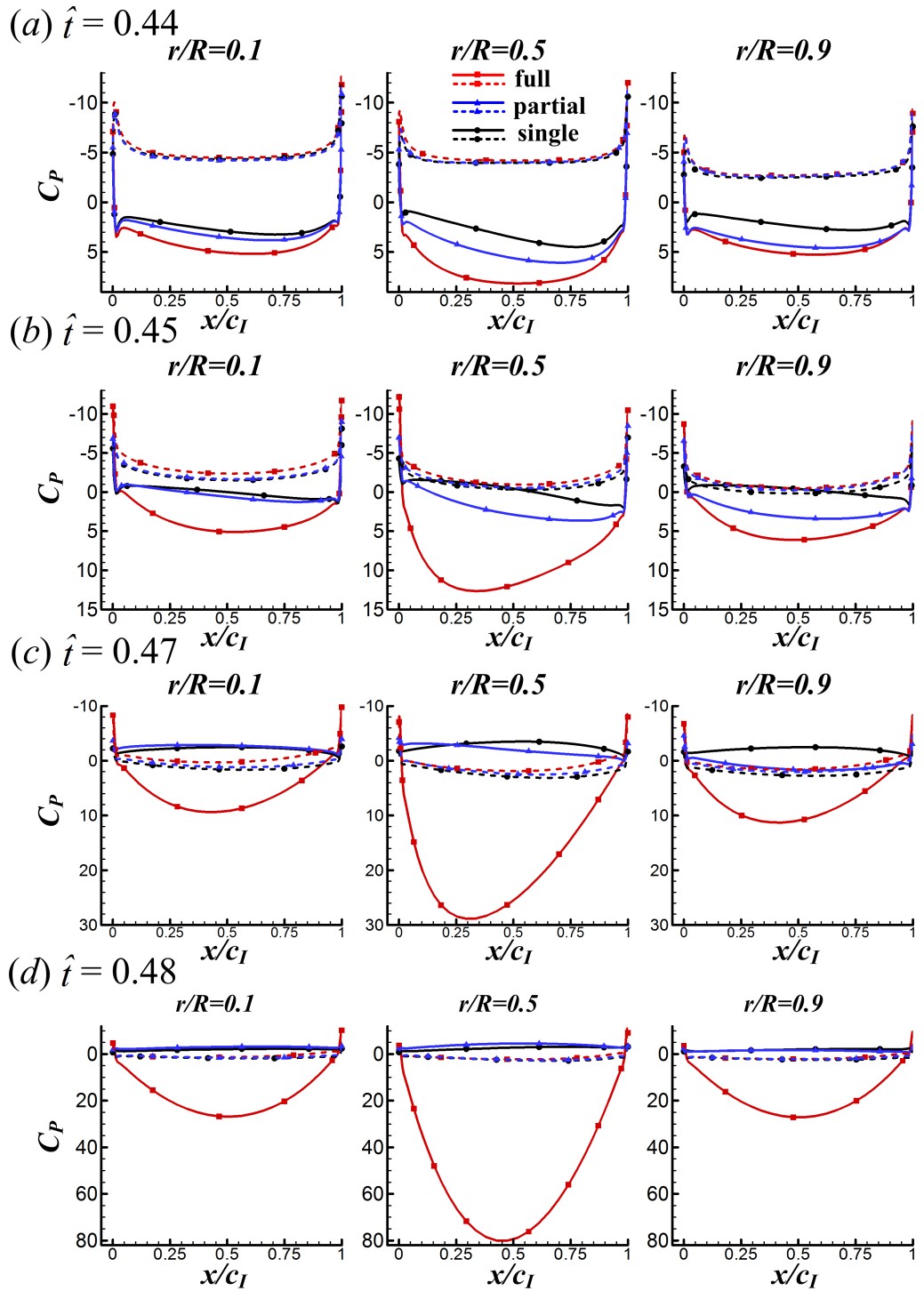

**Figure 8** Surface pressure distributions of the three cases at various spanwise positions for $\hat{t} = 0.44$, 0.45, 0.47 and 0.48 in the clap phase (solid and broken lines indicate the pressure distribution on the lower surface and the upper surface, respectively).

PCF case, the distance between wing roots are much larger than that of wing tips. The wing roots are far apart from each other so that the interaction effect is negligible, the distance between wing tips is almost the same as that of the FCF case and hence the force-enhancing effect. Comparing the $C_P$ distributions of the three cases in Fig. 8 shows that, $C_P$ is almost the same on the upper wing surface (broken lines) while it varies significantly on the lower wing surface (solid lines), indicating that the $C_N$ augmentation in the two-winged cases is directly related to the higher lower-surface $C_P$ (with an exception at $\hat{t} = 0.47$, higher lower-surface $C_P$ of the PCF case results in smaller $C_N$ compared with the SW case, which is discussed later).

Next, how the higher lower-surface $C_P$ is produced by wing interaction is explained and the difference between the FCF and PCF cases is compared. Figure 9 shows the velocity vectors and pressure distributions of the FCF case (A–D), the PCF case (E–H) and the SW case (I–L) in vertical plane at half-wing length at the same four instants as above. In the mid portion of clap ($\hat{t} = 0.44$ and $0.45$), the wings move towards each other quickly, the fluid between the wings is squeezed out of the closing gap and creates a high pressure region. Thus, the positive $C_P$ on the lower surface of wing is greatly increased in the FCF case (Figs. 9A and 9B); because the wings in the PCF case have larger separation distance, the $C_P$ enhancement on the lower surface of wing becomes weaker (Figs. 9E and 9F). In the later portion of clap ($\hat{t} = 0.47$ and $0.48$), the whole wing surfaces in the FCF case are very close to each other, the high $C_P$ region in the very small gap between wings becomes much stronger than before (Figs. 9C and 9D). In the PCF case, due both to the separation between wing roots and rigidity of the model wing (Fig. 4B), only the outer parts of the wings near wing tip are in close proximity, and the rest parts of the wings are still separated largely (Figs. 9G and 9H). Meanwhile, the pitching angular-velocity around the leading edge of wing decreases quickly to near zero at the end of clap phase (see $\dot{\alpha}$ in Fig. 5A). So the interference effect in the PCF case is much weaker and nearly disappears at $\hat{t} = 0.48$ (Fig. 7D).

During $\hat{t} \approx 0.46$–$0.48$, due to the deceleration of pitching angular-velocity (see $\dot{\alpha}$ in Fig. 5A), $C_N$ of the PCF and SW cases becomes negative (see Figs. 5B and 7C) and the absolute value of $C_N$ in the PCF case is even smaller than that of the SW case, bringing benefit of $C_D$ reduction in the PCF case (see Fig. 6C). The reason for this is explained as following. The wing–wing interaction in the PCF case produces a slightly higher lower-surface $C_P$ than that of SW case (see Fig. 8C). As a result, the pressure difference between the lower and the upper surfaces becomes smaller and hence the absolute value of $C_N$ (see Fig. 7C). Because $C_N$ obtained during this short period is manifested mostly as $C_D$, $C_D$ of the PCF case is reduced.

Lower-surface pressure differences ($\Delta C_P$) between the two two-winged cases and the single-winged case at the specified four time instants are shown in Fig. 10, which gives an overall picture of how the interaction effect affects $C_P$ on the whole lower surface of wing. In the FCF case, $\Delta C_P$ is largest at the wing's center of area and decreases form center to border (Figs. 10A–10D); in the PCF case, the $C_P$ enhancement is much smaller compared to the FCF case and reaches its maximum on the wing tip region (Figs. 10E–10H).

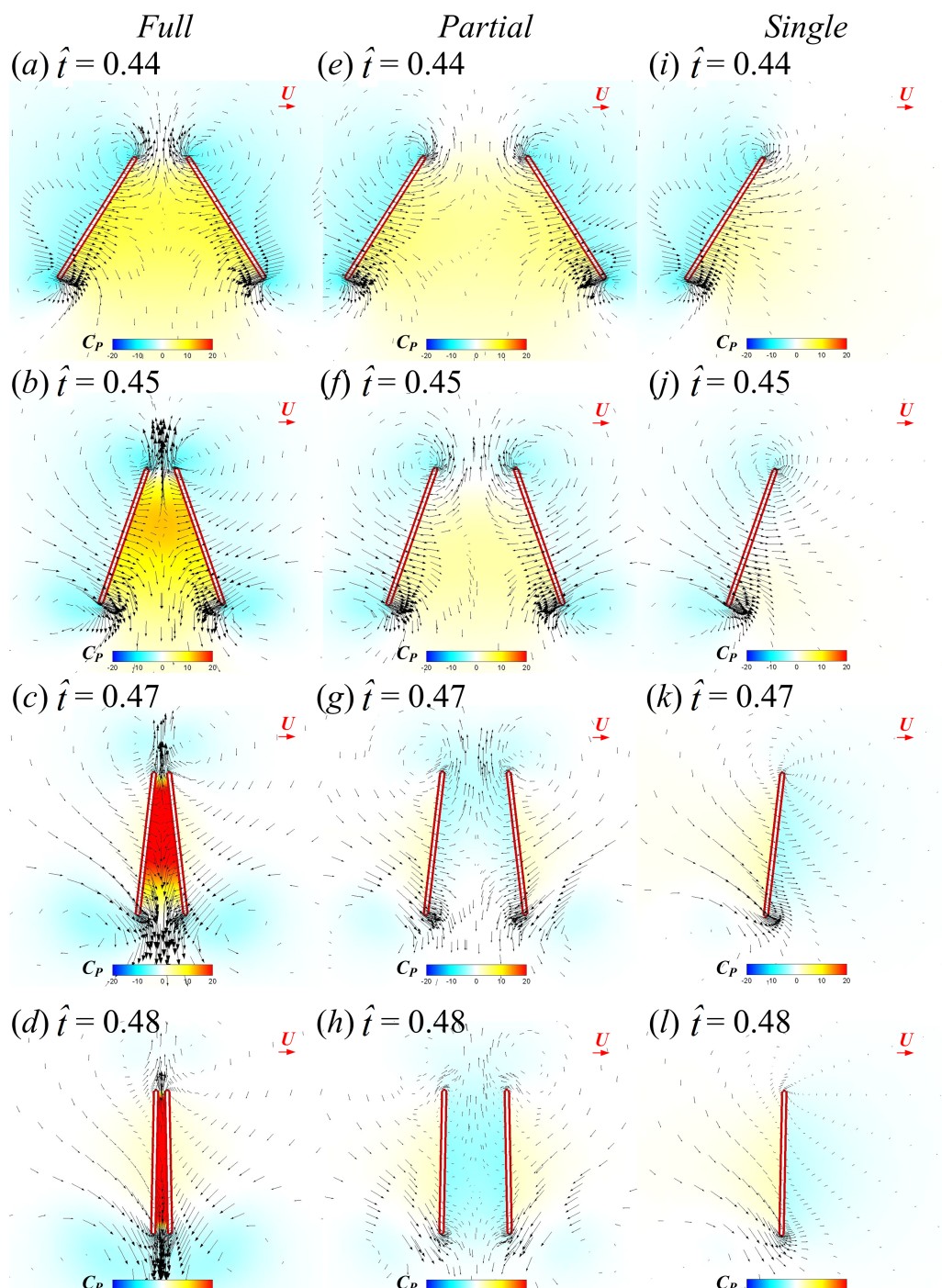

**Figure 9** Velocity vectors and pressure distributions of the FCF case (A–D), PCF case (E–H) and SW case (I–L) in vertical plane at half-wing length for $\hat{t}$ = 0.44, 0.45, 0.47 and 0.48 in the clap phase (red horizontal arrow indicates reference velocity).

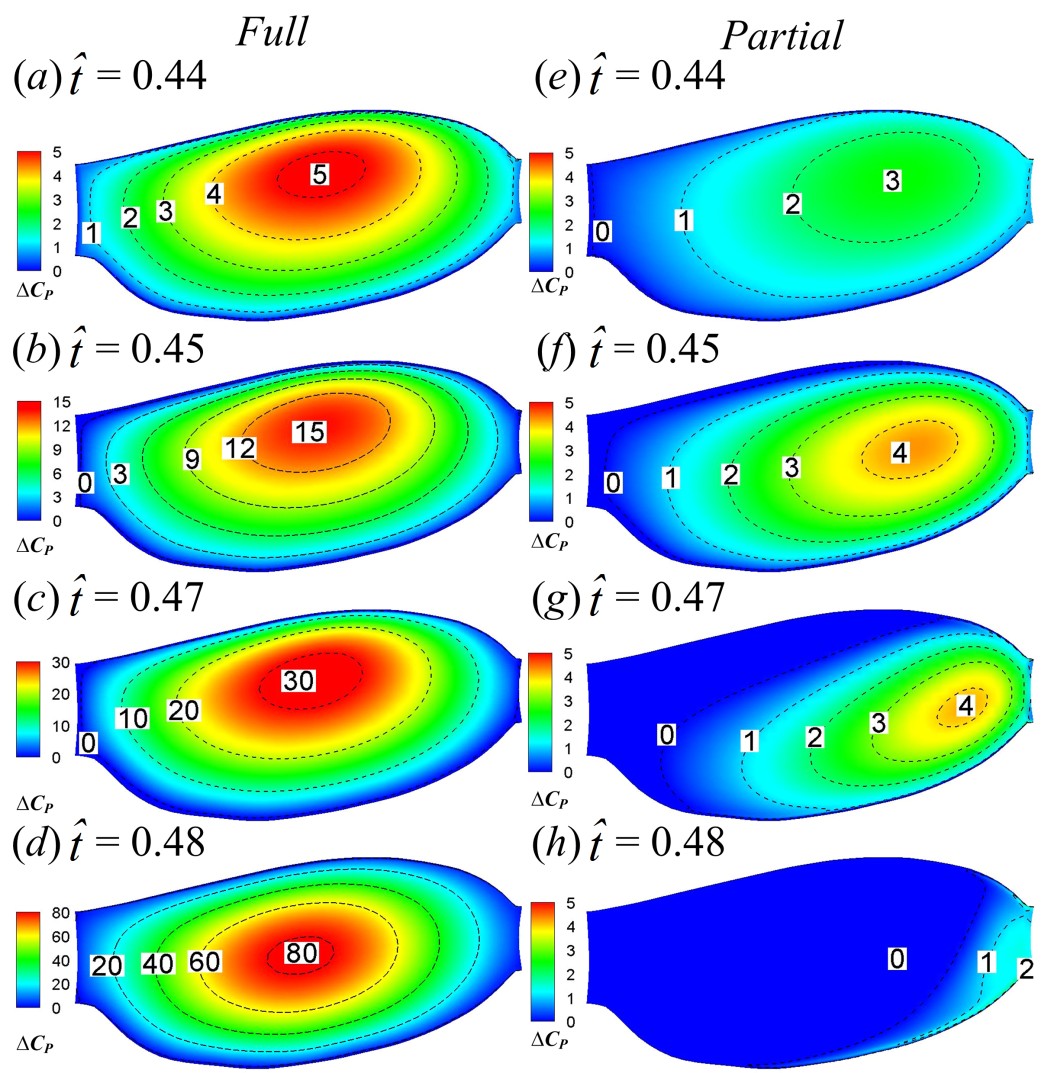

**Figure 10** Lower-surface pressure differences between the two two-winged cases and the single-winged case at $\hat{t} = 0.44, 0.45, 0.47$ and $0.48$ in the clap phase.

## Interference effects between wings in the fling phase

Next, we examine the fling phase ($\hat{t} = 0.50$–$0.61$). Due to the wing–wing interaction, $\Delta C_N$ of the FCF case increases sharply to a peak value ($\hat{t} = 0.52$) immediately after the onset of fling and then decays rapidly in the later fling; in contrast, $\Delta C_N$ of the PCF case varies more gently (Fig. 6A). Figure 11 gives the spanwise distributions of $C_n$ at $\hat{t} = 0.52, 0.53$ and $0.55$ of the three cases. We see that, in the FCF case, when the interaction effect is most obvious ($\hat{t} = 0.52$), the $C_n$ increment is largest at mid-span of wing and decreases gradually toward the sides compared to the SW case (Fig. 11A); later in the downstroke, the $C_n$ increment does not change greatly along the span (Figs. 11B and 11C). In the PCF case, the $C_n$ increment is smaller than that of FCF case all along the wing span, and it is relatively larger in the outer part of the wing ($r$ is large) than that in the inner part of wing ($r$ is small). By comparing the $C_P$ distributions in the three cases in Fig. 12, it is evident

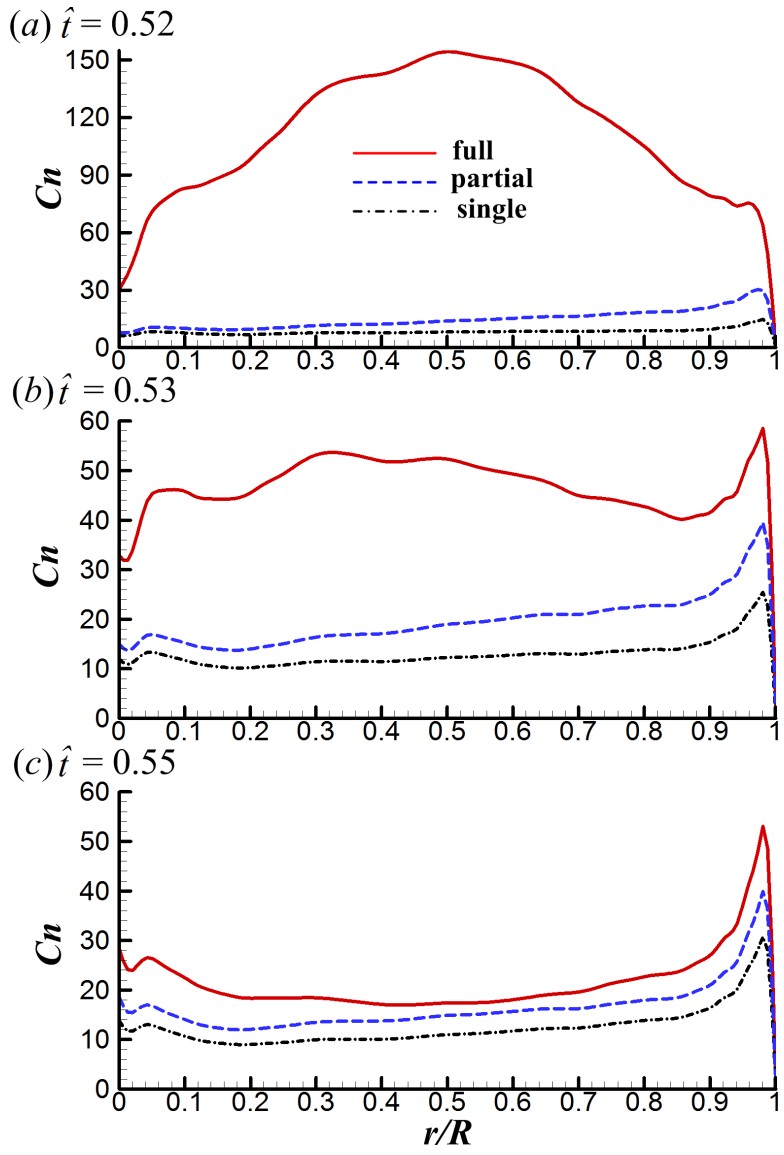

**Figure 11** The spanwise distributions of the sectional normal force of the three cases considered at $\hat{t} = 0.52, 0.53$ and $0.55$.

that the $C_N$ augmentation in the two-winged cases is attributed to the lower upper-surface $C_P$. How the lower upper-surface $C_P$ is produced by wing interaction and the difference between the FCF and PCF cases are discussed as follows. Figure 13 shows the velocity vectors and pressure distributions of the FCF (A–C), PCF (D–F) and SW (G–I) cases in vertical plane at half-wing length. During the fling motion, the wings fling apart about the trailing edge. The gap between the wings is thus expanded and forms a low pressure region, which accounts for the decrease of the upper-surface pressure. At the onset of the fling motion ($\hat{t} = 0.52$ in Fig. 13), the gap between wings in the FCF case is very small and a strong low pressure region is created in the gap when the wing separation occurs (Fig. 13A); in contrast, the gap between wings in the PCF case is large, which weakens

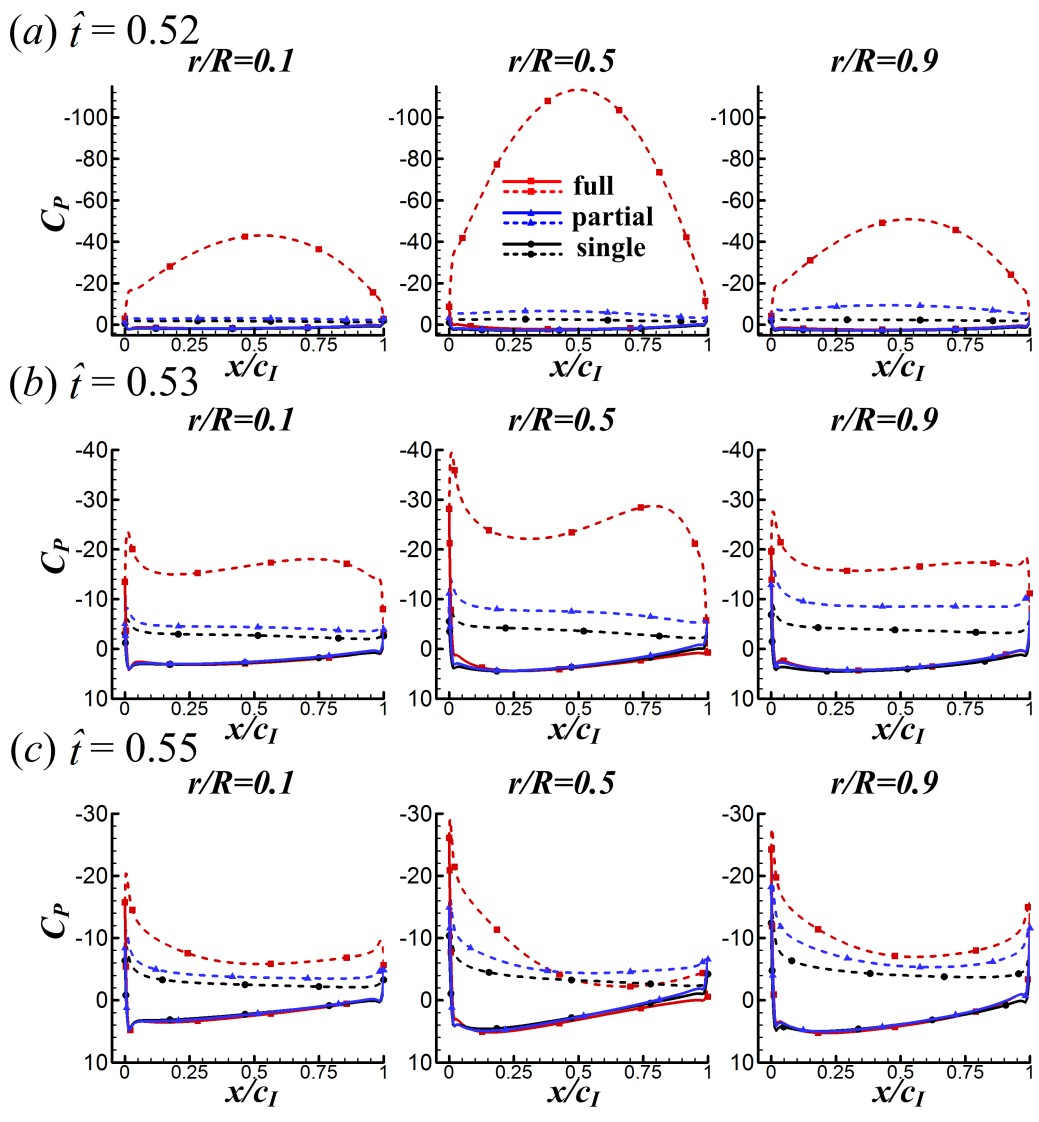

**Figure 12** Surface pressure distributions of the three cases at various spanwise positions for $\hat{t} = 0.52$, 0.53 and 0.55 (solid and broken lines indicate the pressure distribution on the lower surface and the upper surface, respectively).

the effect of wing–wing interaction, resulting in a weak low pressure region in the gap (Fig. 13D). In the later portion of the fling ($\hat{t} = 0.53$, 0.55 in Fig. 13), the low pressure region draws fluid into the opening gap between wings and forms a strong vortex near the leading edge. The leading edge vortex (LEV) creates a low-pressure region and further reduces the amplitude of the negative $C_P$ at leading edge, which explains why the largest negative $C_P$ is obtained at the leading edge at $\hat{t} = 0.53$ and 0.55 (Figs. 12B and 12C). Moreover, an overall picture of how the interaction effect varies on the whole upper surface of wing is shown in Fig. 14, which illustrates the upper-surface pressure differences ($\Delta C_P$) between the two two-winged cases and the single-winged case at several distances during the fling phase. In the FCF case, at the initial start of fling ($\hat{t} = 0.52$), the largest $C_P$

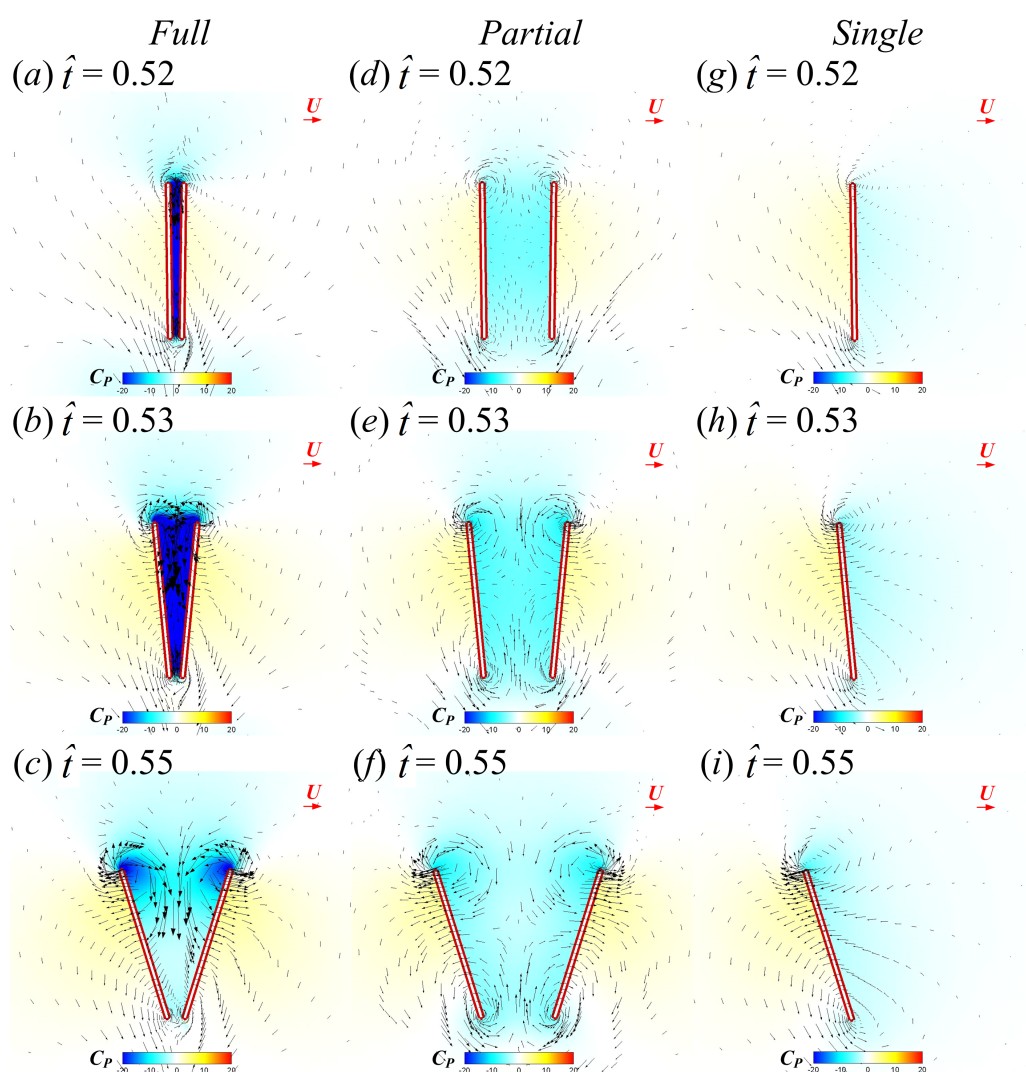

**Figure 13** Velocity vectors and pressure distributions of the FCF case (A–C), PCF case (D–F) and SW case (G–I) in vertical plane at half-wing length for $\hat{t} = 0.52$, 0.53 and 0.55 (red horizontal arrow indicates reference velocity).

decrease is obtained at the wing's center of area (Fig. 14A); once the strong LEV is formed, the largest $C_P$ decrease is obtained at the wing's leading edge (Figs. 14B and 14C). In the PCF case, the amplitude of $C_P$ decrease is smaller and the position of largest $C_P$ decrease also transfers from wing center at $\hat{t} = 0.52$ to leading edge of wing at $\hat{t} = 0.53$ and 0.55, but closer to the wing tip (Figs. 14D–14F).

## Interference effects between wings in the subsequent translation after fling

Though the lift-enhancing effect of wing–wing interaction is mostly restricted to the clap-fling phase, it is still visible in the translational phase after fling (see Fig. 6B). Compared with the SW case, the average $C_L$ during $\hat{t} = 0.61$–1.0 of the FCF case and the PCF case is increased by 20% and 10% respectively. Because lift is increased in the FCF and PCF cases

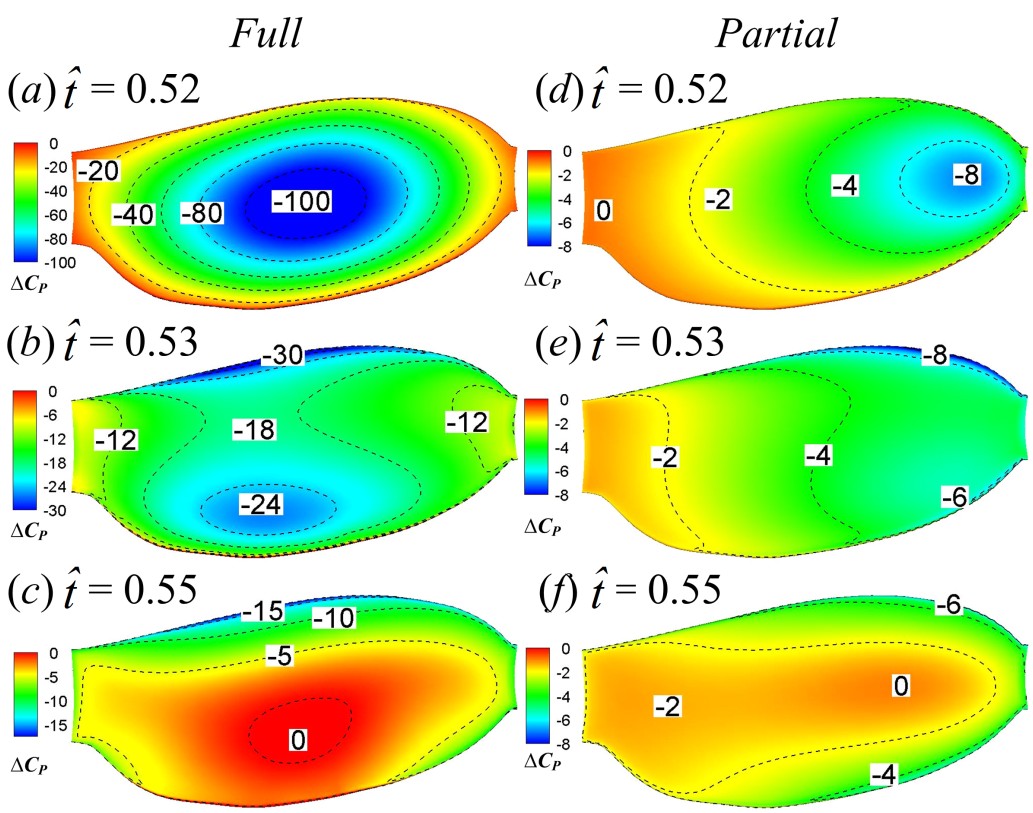

**Figure 14** Upper-surface pressure differences between the two two-winged cases and the single-winged case at $\hat{t} = 0.52, 0.53$ and $0.55$.

for similar reasons and the lift-enhancing effect is more obvious in the FCF case, we only compare the FCF and SW cases. Since the attached LEV is the dominant lift-generating mechanism, the LEV strength may vary between the two cases. Figure 15 shows the is-ovorticity surface plots (top view) and their corresponding spanwise vorticities at the mid span location at $\hat{t} = 0.62, 0.64$ and $0.66$ of the FCF case (A–C) and the SW case (D–F). After the start of fling, a strong $LEV_1$ is generated and then it is shed from the wing in the later part of fling due to the angular deceleration of wing. As the wing continues to rotate, the shed $LEV_1$ peels away from the upper surface of wing and a new $LEV_2$ begins to form and grows quickly (see Fig. 15). Comparing the LEV strength in the FCF and SW cases shows that the strength of both $LEV_1$ and $LEV_2$ in the FCF case is stronger. A possible explanation for this is as follows. A strong low pressure region between wings is created in the two-winged case which sucks more fluid into the opening gap around leading edge and thus creates stronger LEV than that in the single-winged case. Collectively, the lift enhancement after fling in the two-winged cases is attributed partially to the subsequent effect of old LEV generated in the fling phase and partially to the new LEV generated in the translation phase.

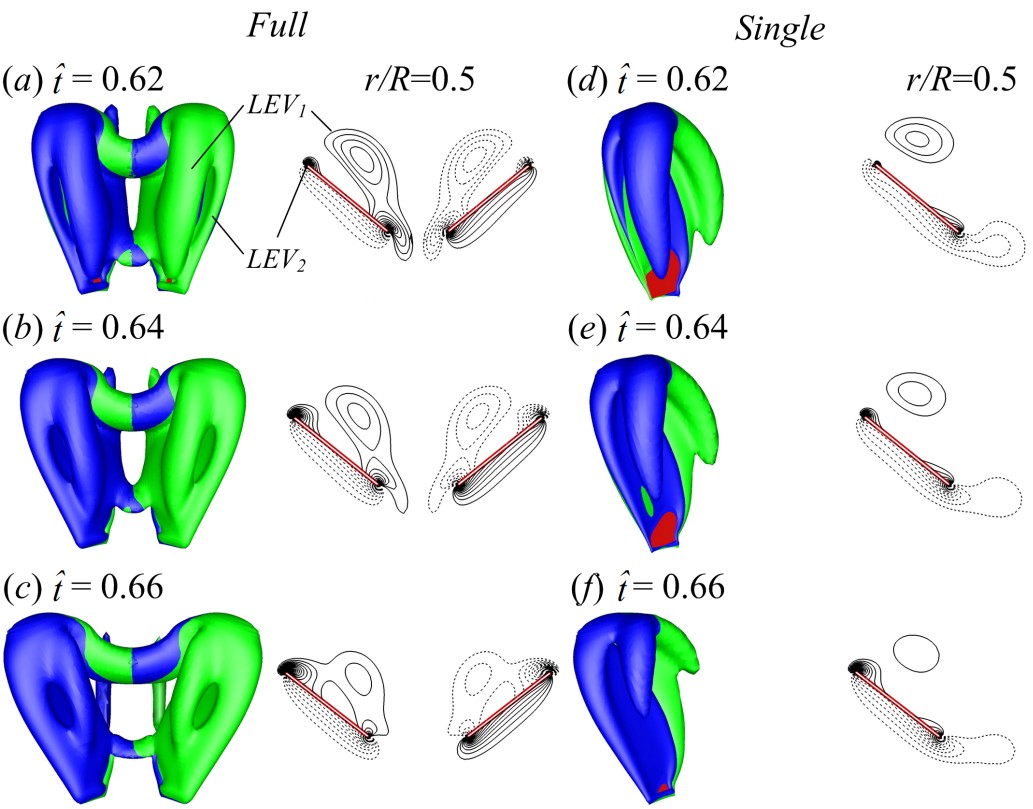

**Figure 15** Iso-vorticity surface plots (top view; the magnitude of the non-dimensional vorticity is 3) and their corresponding spanwise vorticities (the magnitude of the vorticity at the outer contour is 2 and the contour interval is 1) at half-wing length for $\hat{t} = 0.62$, 0.64 and 0.66 of the FCF case (A–C) and SW case (D–F). Vortex structures are shaded by spanwise vorticity to indicate direction: green is negative and blue is positive.

## Calculations in which a sinusoidal variation of angle of attack is employed

Note that in this study, $\alpha$ is assumed to be constant in the translation phase and varies only at stroke reversal. How will the obtained results vary if a different $\alpha$ variation is employed? To see this, we have done some additional computations on a sinusoidal variation of $\alpha$ (see Supplemental Information S2). The results show the following. The interference effect between wings is no longer restricted to the dorsal stroke reversal but extends to the translation phase before and after dorsal stroke reversal, possibly due to the different $\alpha$ variation. Moreover, since the wing root distance is increased, the force enhancement becomes similar among the two-wing cases: $\overline{C_L}$ and $\overline{C_D}$ are increased by about 13% and 8% respectively, compared to the one-winged case.

## Calculations in which other partial clap-fling motions are considered

In the above discussions, only one PCF case with certain root distance and angular excursion is investigated. Since the PCF case, in terms of power efficiency, performs better than the FCF and SW cases, is there a best partial clap-fling motion? To see this, we have done some additional computations in which other root distance and angular excursion combinations

are considered (see Supplemental Information S3). A new variable, the angular distance $\theta$ between the spanwise axis of each wing and the mid plane at $\hat{t} = 0.5$ (the end of the clap phase), is introduced to characterize different motions (Fig. S4A). In all the clap-fling motions, the distance between wing tips at $\hat{t} = 0.5$ is kept constant at $d = 0.1c$. By this definition, the FCF and PCF cases studied correspond to the $\theta = 0°$ case and $\theta = 7.9°$ case respectively. The other four partial clap-fling motions correspond to $\theta = 2°$, $\theta = 4°$, $\theta = 6°$ and $\theta = 10°$, respectively. The results and detailed analysis are given in Supplemental Information S3. The main results are as following. The lift enhancement is largest in the case of $\theta = 0°$ (the FCF case), but the drag is also the largest (i.e., the energy consumption is much larger than that of the SW case). When $\theta$ is increased to $\theta = 2°$, only a slight increase, one can have a large $\overline{C}_L$ augmentation without a large $\overline{C}_D$ ($\overline{C}_L$ is increased by 21% and $\overline{C}_L/\overline{C}_D$ is just slightly lower than that of the SW case). When $\theta$ is further increased, $\overline{C}_L$ and $\overline{C}_D$ continue to decrease gradually while $\overline{C}_L/\overline{C}_D$ stays almost constant. This suggests that the two wings should be close enough, but not too close, to have a good interference effect.

## CONCLUSION

(1) During the clap phase, the wings clap together and create a high pressure region in the closing gap between wings, greatly increasing the positive pressure on the lower surface of wing, while pressure on the upper surface is almost unchanged by the interaction; during the fling phase, the wings fling apart and create a low pressure region in the opening gap between wings, greatly increasing the suction pressure on the upper surface of wing, while pressure on the lower surface is almost unchanged by the interaction.

(2) In the full clap-fling case, the interference effect between wings is most drastic at the end of clap phase and the start of the fling phase: two sharp force peaks (8–9 times larger than that of the single-winged case) are generated. However, as the wing section is nearly vertical, the normal force peaks are mostly manifested as drag and barely as lift of the wing.

(3) In the partial clap-fling case, only the outer parts of wings are in close proximity and the wing separation increases from wing tip to wing root. So the wing–wing interaction effect in the partial clap-fling case is much weaker than that in the full clap-fling case, avoiding the generation of huge drag. The partial clap-fling is a more practical choice for tiny insects to employ: it can augment the mean lift coefficient by about 12% without suffering any efficiency degradation when compared to the single-winged case.

### Funding

This research was supported by a grant from the National Natural Science Foundation of China (11232002). The funders had no role in study design, data collection and analysis, decision to publish, or preparation of the manuscript.

### Grant Disclosures

The following grant information was disclosed by the authors:
National Natural Science Foundation of China: 11232002.

## Competing Interests

The authors declare there are no competing interests.

## Author Contributions

- Xin Cheng and Mao Sun analyzed the data, wrote the paper, prepared figures and/or tables, reviewed drafts of the paper.

## Data Availability

Figshare

https://figshare.com/s/fdcd1e4e7999210b862f

https://figshare.com/s/f6b4f0bed5c4e80c86ff

https://figshare.com/s/bd32c2f11619297697bd.

## Supplemental Information

Supplemental information for this article can be found online at http://dx.doi.org/10.7717/peerj.3002#supplemental-information.

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
