# Peer review of "Aerodynamic forces and flows of the full and partial clap-fling motions in insects"

_PeerJ, doi:10.7717/peerj.3002_

## Round 0.1 · original submission · Minor Revisions

The reviewers all appreciate the current work, but I ask that you specifically revise your manuscript to make clear the differences with your previous work published in Scientific Reports. As suggested by reviewer 1, the work may have greater impact and differentiation if you address the issue of the benefit of partial clap and fling for the single wing scenario.

·

Basic reporting

I think the article is clearly written. The background is relevant to the study and referenced well.The figures are relevant to the article, but it might be better to enlarge the figures 8 and 12 or downsample the vectors (especially around leading and trailing edges) in those figures to show the flow structures and pressure contours clearly at the same time.

Experimental design

The study on the clap and fling of a small insect by the same authors was published in Scientific Reports recently, but this study focuses on the comparison between full and partial clap and fling.
The subject of this study is important in the field of animal flight because the clap and fling is common strategy used by some insects as reviewed in the introduction. However, given that the partial clap and fling is observed in the flight of Liriomyza sativae as reported in the authors’ previous study, I think it would be more meaningful to focus more on the benefit of partial clap and fling in comparison with the single wing case. For example, the drag on the PCF case is even reduced somehow during the clap (figure 5, t^=0.45 – 0.48), which should affect the efficiency. The lift is increased even after the fling (t^>0.55) in the PCF case as well as FCF case as mentioned at lines 305-308. Can the authors explain the aerodynamic mechanisms that cause these changes?

Validity of the findings

The conclusion links to the original question, and is supported by the results. However, I think the original question can be more meaningful as mentioned above.
Grid and time step refinement in the supplementary material is satisfactory. However, it would be nice if the authors can give some discussions about the validity and reliability of the output from their model through the comparison against any other experimental or numerical study. For example, at lines 308-314, the authors summarize the increase in the cycle-averaged lift and drag coefficients. Are these within the similar range of the previous studies reviewed in the introduction?

Additional comments

1) Why does the authors set the minimum distance between wings as 0.1c? Is this based on the measurement in the previous study?

2) Page 6: “For wings flapping at low Re, the resultant aerodynamic force is almost normal to the wing surface and …”
It might be better to reword this sentence like “For the flapping wing in this study, the Re is still high enough to assume that the aerodynamic force is almost normal to the wing surface”, since the viscous force becomes more important when the Re is lower.

Reviewer 2 ·

Basic reporting

This is a well paced and a well written manuscript. It compares the aerodynamic characteristics of the full vs partial "clap-fling" mechanism in small insects. The clap & fling aerodynamic mechanism was investigated in many previous studies including a recent study by the authors and other studies by the second author. The contribution of the paper is in the detailed simulations (particularly pressure distributions) for both full and partial clap-fling motions.

If possible, I think that the study will benefit from a list of symbols.

Recommendations to improve figures:

1) In all figures: Please be consistent in the symbols font, symbols size and symbols orientation (i.e. italic/normal).

2) Figure 2(b): the y-axis should be labelled and it is better to use degrees rather than radians for improved clarity.

3) Figures 7 and 11: Please be consistent in using line types; i.e. solid always for lower surface or vice versa.

4) Figures 7 and 11: What is the subscript used for c?

5) Figures 9 and 13: In each of these figures, it will be more appropriate to unify the colour map scale so that the huge differences between the three time instances are clearly demonstrated.

Experimental design

Specific comments on the design of the study:

1) An important part of this study is the kinematic patterns assumed, in particular the AoA variation. Given that the study is not constrained by kinematic patterns measured for a certain insect, how will the obtained results vary if a different AoA variation (e.g. sinusoidal variation) is employed? A demonstration of this point should be useful to the reader. Related to this point, the AoA variation at the beginning and the end of stroke is different from where the clap-fling happens, why is this? A comment would be useful.

2) The wing geometrical parameters are taken from [14] whereas the kinematic parameters (page 10) are from [35], why is there no consistency in the simulation inputs?

3) The definition employed for partial clap-fling seems arbitrary. How will the results get affected if different values for the gap distance between wings and different angular excursions are simulated? A sensitivity analysis of these two parameters should be useful to define the limits between what we can consider as full and what we can consider as partial. Also, it would be useful to present these variables as non-dimensional ratios, e.g. the gap distance between wings to be fraction of wing length R.

Validity of the findings

The results appear to be sound, only few points to ask:

1) The authors provided some details of the numerical solver in the supplementary materials; however, it is a typical practice within CFD studies to provide some validation of the numerical solver before going to extract findings. It is recommended to add a useful validation against previous clap-fling studies.

2) Figures 4 and 5: There are 2 peaks appearing in the clap phase. Can the authors explain more on this behaviour?

3) Line 432: This is the first mention of LEV. There was no mention of the LEV within the clap analysis. A comment may be useful.

Additional comments

Comments to improve clarity:

1) Page 2: Line 31: Abstract: the statement "relevant to the smallest flying insects": I would be more conservative when using the word "smallest". There are insects that can be considered smaller than the ones considered here, e.g. see studies on Thrips with comb/bristled wings.

2) Page 4-6: Lines 84-125: This is a very long paragraph.

3) Lines 213-215: Not quite clear what the authors are trying to say; can you clarify more?

4) May be I have missed this, but which flapping cycle is simulated? Or is it just one flapping cycle?

5) It may be useful to demonstrate the motions of the wing pairs (as in Fig 3) for the complete cycle.

6) Lines 225-227 are essentially the same as lines 234-237. There is no need for excessive repetitions.

7) Line 239: "perpendicular to wing span" not very accurate as the meaning can be interpreted in different ways. Consider removing.

8) It is recommended to avoid the usage of the word "let us".

9) Line 358: Not quite clear what is the point of this sentence. Please clarify

Reviewer 3 ·

Basic reporting

No comments

Experimental design

No comments

Validity of the findings

No comments

Additional comments

The paper reports an investigation on the different effects of full and partial clap-fling motions in aerodynamic force generation at low Reynolds number using numerical simulation. It was found the partial clap-fling motion, in terms of power efficiency, is more practical choice for tiny insects to employ increase lift.
The method used in the paper is suitable for the study, and the paper is well written. I recommend that this paper be accepted but only after a minor revision, and subject to the specific comments below being addressed.

Comments:
1. Line 51 and 52, “Average-sized insects”. The authors mentioned that average-sized insects have a wing length about 3-4mm, but the examples given, i.e. refs 2-8, are mostly on much larger insects. I think the term “Average-sized insects” should be replaced by “tiny insects”.

2. Line 51, ref. 1 is a book. It will be easier for readers if the author can indicate the specific chapter and section or page number.

3. The authors may wish to also consider the following references on the aerodynamics of rigid and flexible Hawkmoth wing:

Lua KB, Lee YJ, Lim TT and Yeo KS 2016 Aerodynamic effects of elevating motion on hovering rigid hawkmoth-like wings AIAA J 54 2247-2264.

Lu H, Lua KB, Lim TT and Yeo KS 2016 Ground effect on the aerodynamics of three-dimensional hovering wings Bioinspiration & Biomimetics 11.

Lua KB, Lai KC, Lim TT and Yeo KS 2010 On the aerodynamic characteristics of hovering rigid and flexible hawkmoth-like wings. Experiments in Fluids 49 1263-1291.

4. Line 140, rigid wing is used in this study. It is well known that insect wings are highly flexible and deformed significantly under the aerodynamic and inertia forces. In the case of clap and fling, the flexibility effects may be even more significant as the wing deformation will change the distance between and thus the interaction of the two wings. Can the author provide justification on using rigid wings to study clap and fling effects?

5. Line 143, The authors select the mean velocity at the wing radius of gyration as the reference velocity. This decision is supported by the finding of “Lua KB, Lim TT and Yeo KS 2014 Scaling of aerodynamic forces of three-dimensional flapping wings AIAA J. 52 1095–101”. I suggest the authors to cite this paper.

6. Line 209, How did the author decide on the current root distance and angular excursion? Since the PCF performs better than FCF and SW, there should be a best PCF motion, why the authors did not try other root distance and angular excursion combinations?

7. Figure 5 (c) when t^=0.46 the difference of drag coefficient for PCF motion is lower than 0, meaning that at this point of time the PCF motion can reduce drag. Can the author explain more details about this reduction in drag?

8. Figure 8 & 12 only show the flow fields of full and partial clap-fling motions. It will be clearer if the authors can show and compare the results of single wing motion in the figures as well.

---

## Round 0.2 · accepted · Accept

The referees have a couple of very minor grammatical revisions. Please include these into your manuscript as soon as possible when contacted by the production team.

·

Basic reporting

no comment

Experimental design

no comment

Validity of the findings

no comment

Additional comments

lines 301, 376, 411,415: Alpha_dot is not printed properly.

lines 534-554: For consistency, it is better to use the root distance rather than the angular distance, or to give the root distances for all of the angular distances (2,4,6 and 10 degrees in addition to 0 and 7.9 degrees).

Reviewer 2 ·

Basic reporting

None

Experimental design

None

Validity of the findings

None

Additional comments

The authors have done a good job in addressing the comments from reviewers. If possible, the following very minor points would be useful to consider during the production process:
1) Lines 49-50: Added lines can be more attractive, and require grammatical corrections. Try to improve.
2) Figures 10 & 14: Colour bar scale is not unified as requested. Is it possible to have the same upper and lower bound values for delta_Cp applied to all subplots?
3) Added text between lines 521-529 and lines 530-550: To highlight the relevance of these two discussions, it may be better to include subheadings for these two discussions.

Reviewer 3 ·

Basic reporting

I have no further comment to the authors.

Experimental design

I have no further comment to the authors.

Validity of the findings

I have no further comment to the authors.